

# Diversity and efficacy of insect sampling methods in an urban tropical dry forest of the Colombian Caribbean

Rodrigo Sarmiento-Garcés,  Daniel Posada-Echeverría and  Rafik Neme

Departamento de Quimica y Biologia, Universidad del Norte, Barranquilla, Atlántico, Colombia

## ABSTRACT

Tropical dry forests (TDFs) are among the most threatened ecosystems in Colombia and globally. Characterized by distinct dry and wet seasons, TDFs support a rich diversity of flora and fauna and offer crucial ecosystem services, despite facing significant threats such as deforestation, urbanization and climate change. This study provides a comprehensive assessment of insect diversity within an urban TDF area in Puerto Colombia, Atlántico and in the Colombian Caribbean, by comparing three traditional sampling methods: Malaise traps, pitfall traps, and canopy fogging. In total, 3,135 insects were collected, representing 413 morphospecies across various trophic guilds. Our findings indicate that Malaise traps were the most effective in capturing a diverse range of species, followed by pitfall traps, while canopy fogging yielded the lowest species richness. Each method captured unique species, demonstrating that these methods are complementary in capturing the full spectrum of insect diversity in TDFs. Additionally, our study contributes valuable data on the taxonomic and functional diversity of insects in this continuously threatened ecosystem, underscoring the need for conservation efforts in urban and peri-urban TDFs.

## INTRODUCTION

Tropical dry forests (TDF) occur in dryland environments in tropical regions. In the neotropics, they are defined as a biome which encompasses a set of floristic groups, with a wide distribution, ranging from Mexico to Argentina (*DRYFLOR et al., 2016*). TDFs hold significant value due to their strong seasonality in precipitation and marked heterogeneity in rainfall seasonality, soil characteristics, frost, and altitude (*González et al., 2018*). TDFs are vital for biodiversity conservation and provide critical ecosystem services, including carbon storage, climate regulation, nutrient cycling, and economic and recreational values (*Lewis et al., 2009*; *Siyum, 2020*; *Vinya et al., 2019*; *Zhou et al., 2013*). However, they are among the most threatened ecosystems globally, suffering from deforestation, land-use change and climate change, making them one of the most threatened and impacted tropical terrestrial ecosystem (*Ferrer-Paris et al., 2019*; *Miles et al., 2006*).

In the Colombian Caribbean, TDFs have been greatly impacted by human activities as a consequence of fragments being located on lands with the most fertile soils, which leads to the promotion of productive activities, leading to an increase of land-use change

Corresponding author
Rafik Neme, rneme@uninorte.edu.co

for agriculture, and urbanization (*Correa Ayram et al., 2020*). The remaining dry forest patches are highly fragmented, leading to the isolation of plant and animal populations, and reducing their resilience to environmental stressors (*Pizano & García, 2014*).

Despite the singularity, importance, and degree of threat to neotropical dry forests, studies on them are scarce and limited to a few biotic communities (*IAvH, 1998*; *Pizano & García, 2014*). In the case of insects, research has primarily focused on specific taxa, with limited studies covering the entire terrestrial insect community or a broad spectrum of trophic levels (*Aldana-Domínguez et al., 2017*).

Insects play critical roles in TDFs, participating in processes like pollination, nutrient cycling, decomposition, and providing ecosystem services, including pollination and pest control. Understanding the diversity, community structure, and trophic guilds of insects in these forests is crucial for effective conservation measures. There is a need for more comprehensive research on the use of various types of traps, especially those capturing a wide variety of insect species, such as Malaise or pitfall traps, or in less studied regions like Colombia (*Hausmann et al., 2020*).

Various insect sampling methods, each with their own limitations and biases, are essential tools for studying insect diversity and community structure, especially in the face of ongoing challenges such as mass extinctions and climate change. While each method has its strengths, employing multiple approaches—such as Malaise traps, pitfall traps, and canopy fumigation (fogging)—provides a more comprehensive understanding of insect diversity and community dynamics in complex ecosystems. Malaise traps, in particular, have proven effective in capturing a broad range of flying insects, offering valuable insights into the diversity and trophic structure of insect communities in TDFs (*Borkent et al., 2018*; *Hausmann et al., 2020*). This study aims to assess the diversity and trophic guilds within a tropical dry forest in Puerto Colombia, while critically evaluating the effectiveness of these three widely used sampling methods. By integrating these methods, we seek to contribute a detailed understanding of taxonomic and functional insect diversity in this threatened ecosystem, highlighting the importance of using complementary sampling strategies to capture the full spectrum of biodiversity.

## METHODS

### Area of study

Our study was conducted in the municipality of Puerto Colombia, Atlántico, part of the Metropolitan Area of Barranquilla, in the Colombian Caribbean Region. The area of interest is in the Pericaribbean Arid Belt (*Hernández et al., 1992*), with patches of tropical dry forest (TDF) (*Holdridge et al., 1971*; *Sánchez-Azofeifa et al., 2005*). It has an altitude of 4 m above sea level, an approximate rainfall of 900 mm, an average temperature of 27 °C, relative humidity ranging between 75–85%, and evaporation rates of 1,500 to 1,900 mm (*IDEAM, 2020*).

The sampling was carried out in a secondary tropical dry forest inside Universidad del Norte in November 2019 with an approximate area of 8.5 Ha, surrounded by a mosaic of developing urban landscapes dominated by secondary vegetation. The forest presents a

well-defined stratification with a low or herbaceous stratum dominated mainly by herbs and creeping lianas, a medium or shrubby stratum composed of shrubs or small trees (diameter at breast height (DBH) $\leq$ 5 cm), and a canopy or tree stratum made up of larger trees (DBH $\geq$ 5 cm) reaching heights of up to approximately 12 m.

## Sampling of terrestrial entomofauna

Different collection methods were used to obtain a representative sample of the available entomofauna in the forest described above. The effectiveness of the three collection methods was evaluated, and the contribution and complementarity of each method to the global collection were assessed. The methods used were selected to sample each of the strata present in the forest. For this purpose, pitfall traps were used for the low stratum, Malaise-type flight interception traps for the medium or shrubby stratum, and the fumigation method of the canopy or fogging for collections in the high or arboreal stratum. The collections were carried out in October 2019 during the season of humidity and maximum temperature values. Below is a brief description of the collection methods used.

## Pitfall traps

To capture terrestrial insects present on the soil surface, a transect of 12 pitfall traps was set up, with 20 m between each trap. The activity period for each trap was eight days, resulting in a total exposure time of 192 h per trap. Each trap consisted of a plastic container (25 cm in diameter, 25 cm deep), placed at ground level without any attractant, and half-filled with a mixture of water, detergent, and 96% alcohol to maximize sample preservation. A plastic cover, supported by wooden sticks at approximately 15 cm above the trap, was used to minimize water entry from rain, and the container had sufficient empty volume to prevent spillage.

## Malaise traps

In the case of insects associated with the herbaceous and shrubby or medium stratum, four Malaise-type flight interceptor traps were installed, separated by 30 to 50 m between each trap. The installed traps consisted of the classic Malaise traps with front height dimensions of 1.80 m, rear height of 1.10 m, and a length of 1.60 m. The collection devices were active for eight days (192 h per trap).

## Canopy fogging

For the canopy stratum, an area of 30 m2 was fumigated with broad-spectrum pyrethroid insecticide (Pyrethrins 3% w/v + Piperonyl Butoxide 15% w/v EW), from which four independent samples of 4.2 m2 were taken (1.2 m $\times$ 3.5 m). The collection exposure time after fumigation was two hours, during which the insecticide had a residual effect.

All the collection material was labeled and preserved in 96% alcohol until further processing.

## Taxonomic identification

Once the material was collected from each of the methods, taxonomic determination and quantification were carried out. For each collection method, all the organisms were separated at the morphospecies level and identified to the lowest possible taxonomic level

that would allow the identification of the trophic guild. The identified morphospecies were counted to establish their abundances. The keys and diagnosis of *González & Carrejo (1992)*, *White (1983)*, *Arnett Jr & Thomas (2000)*, *Triplehorn, Johnson & Borror (2006)*, *Fernández & Sharkey (2006)* were used for the taxonomic determination at the family and superfamily level.

Due to the prolonged exposure of Malaise traps for eight days, the captured Lepidoptera were in a deteriorated state. These specimens were excluded from the study to avoid potential biases in estimating species diversity. Additionally, since our study was specifically focused on the class Insecta, arachnids captured by the sampling methods were not included in the analysis.

## Allocation of trophic guilds

For the characterization of the trophic groups of each taxon, a review of the literature on their life histories and eating habits was carried out based on the literature indicated for taxonomic identification. Thus, 11 categories or trophic guilds were established: omnivorous, parasitoids, mycophages, saprophagous/coprophagous, nectarivorous/polynivorous, gall-forming phytophagous, chewing phytophagous, sap-sucking phytophagous, xylophagous/stem-boring/seed-eating phytophagous, predators, and hematophagous. This classification was based on the definition of functional groups by *Steneck (2000)* and *Clavijo-Awazacko & Amarillo-Suárez (2013)*, who define them as a group of species that play equivalent functional roles in communities and ecosystems.

## Data analysis

The insect communities present in each collection method were compared using confidence intervals (95%) (Chao1 estimator) of the rarefaction curves and extrapolation of the Hill numbers, using the richness ($q = 0$) and diversity values ($q = 1$ and $q = 2$) (*Chao et al., 2014*; *Hsieh, Ma & Chao, 2016*), with a bootstrap of 999 iterations. Likewise, through these estimators, the percentage of sampling coverage was evaluated for each of the methods used, as well as for each of the orders of insects collected, evaluating the collection methods.

For the comparison of the composition between communities according to each collection method (beta diversity), we used a multivariate analysis of variance (Permutational Multivariate Analysis of Variance (PERMANOVA)) using the adonis function with 999 permutations in the vegan package (*Oksanen et al., 2017*). With the betapart package (*Baselga & Orme, 2012*), diversity was partitioned into nestedness and turnover to understand if the result of dissimilarity is due to substitution or loss of species between the different collection methods.

For the analysis of trophic guilds, a one-way Analysis of Variance (ANOVA) was performed to compare the richness of trophic guilds for each collection method. Additionally, the Equity index J was estimated to evaluate the distribution of richness and understand if there is dominance by a few guilds in some of the methods.

All statistical analyses were performed using the statistical environment R (*R Core Team, 2023*).
**Table 1** **Richness of insect trophic guilds and their equitativity.** Richness and equity index J for three different collection methods, as a proxy for functional diversity distribution across methods.

|  | Pitfall | Malaise | Fogging |
|---|---|---|---|
| Trophic guilds present | 9 | 11 | 9 |
| Equitativity J | 0.89 | 0.84 | 0.95 |

## RESULTS

A total of 3,135 individuals of the class Insecta, distributed among 413 morphospecies, were collected. The orders with the highest representation were Hymenoptera (137 sp, 1,280 individuals), Coleoptera (125 sp, 1,237 individuals), Diptera (87 sp, 377 individuals), and Hemiptera (36 sp, 120 individuals), which represent about 95% of the total reported richness. The other orders had considerably lower presence with fewer than nine species (Table 1, Fig. 1).

Of all the species captured, the Malaise trap was the most effective capture method (284 sp, 1,174 individuals), presenting more than twice the species richness than the second most effective method (104 sp, 1,127 individuals), the pitfall traps. Finally, canopy fogging was the method that presented the lowest richness values (78 sp, 834 individuals), presenting about a third of the richness collected by the Malaise method.

The Malaise trap was mainly represented by the orders Hymenoptera, Diptera, and Coleoptera, which accounted for 91% of the species richness collected for this method. The pitfall traps mainly collected organisms from the orders Coleoptera and Hymenoptera (mainly ants) with 72% of the species present, while the canopy fogging method was represented mostly by the orders Hymenoptera, Coleoptera, and Hemiptera with 82% of the species (Fig. 2).

The sampling coverage was adequate for each of the collection methods, where the canopy fogging and pitfall trap methods had the best performance, collecting about 95% of the potential species compared to the Malaise trap method, which had a coverage percentage of 85% (Fig. 3). However, despite the good results found in sampling coverage, when insect orders are analyzed separately, we see a somewhat different picture, where orders such as Mantodea and Orthoptera present percentages of sampling coverage below 60%. The remaining orders had a representation greater than 80% coverage percentage (Fig. S1). Orders such as Collembola, Embioptera, Neuroptera, and Zygentoma were not included in this analysis due to their low representativeness with abundances less than five individuals, which denotes the low effectiveness of these methods for the collection of the orders.

The rarefaction curves showed significant differences in richness for each of the methods explored, with non-overlapping confidence intervals. The Malaise trap presented close to 70% of the total collected richness, followed by the pitfall traps with 25% of the richness, and the canopy spraying method with 20%. Among the used methods, our result show us that Malaise traps are the single most effective mechanism for sampling insect communities, with the ability to capture almost three times more richness than the other two methods compared (Fig. 4A).

| Order | Total Richness | Pitfall | Malaise | Fogging |
|---|---|---|---|---|
| Hymenoptera | 137 | 29 → | 102 ↑ | 25 → |
| Coleoptera | 125 | 47 ↑ | 77 ↑ | 25 → |
| Diptera | 87 | 9 → | 80 ↑ | 2 ↓ |
| Hemiptera | 36 | 7 → | 18 ↑ | 14 ↑ |
| "Psocoptera" | 9 | - ✕ | 2 → | 9 ↑ |
| Blattodea | 4 | 4 → | 1 ↓ | - ✕ |
| Mantoidea | 5 | 1 ↓ | 1 ↓ | 3 ↑ |
| Orthoptera | 5 | 3 → | 2 ↓ | - ✕ |
| Collembola | 2 | 2 ↓ | - ✕ | - ✕ |
| Embioptera | 1 | 1 ↓ | - ✕ | - ✕ |
| Neuroptera | 1 | - ✕ | 1 ↓ | - ✕ |
| Zygentoma | 1 | 1 ↓ | - ✕ | - ✕ |

| | Species count | Sample Coverage | Total richness |
|---|---|---|---|
| ↑ | > 10 | > 80 % | > 50 % |
| → | ≥ 4 | > 50 % | > 30 % |
| ↓ | < 3 | occasional sample | |
| ✕ | No reports | | |

**Figure 1** **Richness by orders of insects and efficiency of three different collection methods and their efficiency in a relict of dry forest in the municipality of Puerto Colombia–Atlántico.** Icons indicate efficiency and richness ranks for each order and each sampling method. Silhouettes illustrate common body plans for each order. Silhouette sources: Hymenoptera icon, CC0 1.0, https://commons.wikimedia.org/wiki/File:Hymenoptera_icon.svg; Coleoptera icon, CC0 1.0, https://en.m.wikipedia.org/wiki/File:Coleoptera_icon.svg; Diptera_icon, CC0 1.0, https://commons.wikimedia.org/wiki/File:Diptera_icon.svg; Hemiptera icon, CC0 1.0, https://commons.wikimedia.org/wiki/File:Hemiptera_icon.svg; Psocoptera icon, CC0 1.0, https://commons.wikimedia.org/wiki/File:Psocoptera_icon.svg; Grasshopper, CC0, https://www.rawpixel.com/image/9069335/vector-cartoon-illustrations-animal; Mantis religiosa, CC0 1.0, https://www.phylopic.org/images/c5a14ad6-6d6f-49db-97b9-680cb36f8646/mantis-religiosa; Blattella germanica, CC0 1.0, (continued on next page…)

Diversity analysis, assuming relative abundance, showed similar results to those obtained in richness analyses, where the Malaise trap presents the highest diversity values for both typical species (Shannon, $q = 1$) and abundant species (Simpson, $q = 2$) (Figs. 4B and 4C).

For each type of trap, we can see that the typical species (Shannon, $q = 1$) in the Malaise trap occupy about 42% of the richness, while the abundant species (Simpson, $q = 2$) are 20% of all the species. The pitfall traps had a percentage of 20% of typical species, while the abundant species were 11%. The canopy fogging method reported 24% typical species and 15% abundant species. The previous results show us that, despite the high efficiency of the traps, the collected communities are dominated by a low number of species, which tells us of a low equality in the distribution of abundance.

Among the unique features of each trap (Figs. 5A–5C), we can see that the canopy fogging method provides 51 species represented in the families Buprestidae, Cerambycidae, Chrysomelidae, Cicadellidae, Coccinellidae, Corylophidae, Curculionidae, Dermestidae, Dictyopharidae, Formicidae, Histeridae, Lampyridae, Mantidae, Membracidae, Miridae, Ploiariidae, Reduvidae, Thespidae and Tingidae. Further, we observed higher richness of Psocoptera compared with the other traps.

Pitfall traps presented 74 exclusive species represented in the families Acanthopodidae, Aphodiinae, Aradidae, Blattelidae, Blattidae, Bolboceratinae, Ceratocanthidae, Chrysomellidae, Cicadellidae, Curculionidae, Elateridae, Endomychidae, Formicidae, Gryllidae, Histeridae, Isotomidae, Lygaeidae, Muscidae, Mutillidae, Mycetophilidae, Nicoletiidae, Phoridae, Pompilidae, Pyrrhocoridae, Scarabaeidae, Silvanidae, Staphylinidae, and Tenebrionidae.

Malaise traps presented 241 exclusive species represented in the families Agromyzidae, Ampulicidae, Anisopodidae, Anobiidae, Anthicidae, Anthomyiidae, Asilidae, Bostrichidae, Braconidae, Calliphoridae, Canacidae, Cecidomyiidae, Cerambycidae, Ceratopogonidae, Cercopidae, Chalcidoidea, Chamaemyiidae, Chironomidae, Chloropidae, Chrysomelidae, Cicadellidae, Cleridae, Clusiidae and Cyphotidae.

In terms of the composition of the species and their abundance (Bray–Curtis dissimilarity), we found that there were significant differences between each of the collection methods ($F = 20.10$; $P < 0.001$). This shows us, that in addition to the differences in richness and diversity between methods, each of the methods collects a particular composition of fauna. Now, it is important to understand whether the observed dissimilarity can be explained as nested behavior (a sampling type is a subset of another) or turnover of species between methods (a method captures species that another cannot).

The results show, first of all, that there is a high dissimilarity between sampling methods (total beta diversity (SOR) = 0.88), which is given predominantly by species turnover (turnover-driven beta diversity (SIM) = 0.77; 87%) with a lesser contribution from
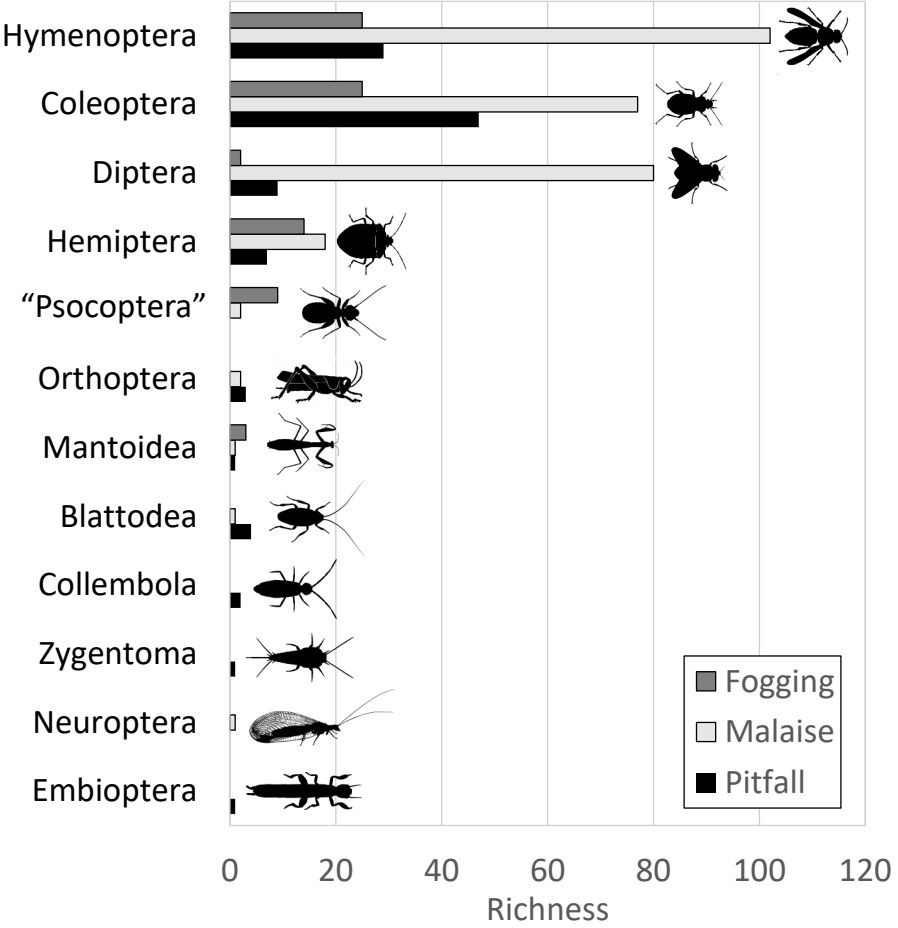

**Figure 2** **Distribution of abundance by each order of insects in each of the collection methods evaluated.** Each bar indicates the number of morphospecies (observed richness) for each order. Different colors in each order indicate richness by type of trap. Silhouettes illustrate common body plans for each order. Silhouette sources: Hymenoptera icon, CC0 1.0, https://commons.wikimedia.org/wiki/File:Hymenoptera_icon.svg; Coleoptera icon, CC0 1.0, https://en.m.wikipedia.org/wiki/File:Coleoptera_icon.svg; Diptera_icon, CC0 1.0, https://commons.wikimedia.org/wiki/File:Diptera_icon.svg; Hemiptera icon, CC0 1.0, https://commons.wikimedia.org/wiki/File:Hemiptera_icon.svg; Psocoptera icon, CC0 1.0, https://commons.wikimedia.org/wiki/File:Psocoptera_icon.svg; Grasshopper, CC0, https://www.rawpixel.com/image/9069335/vector-cartoon-illustrations-animal; Mantis religiosa, CC0 1.0, https://www.phylopic.org/images/c5a14ad6-6d6f-49db-97b9-680cb36f8646/mantis-religiosa; Blattella germanica, CC0 1.0, https://www.phylopic.org/images/b2e26249-80be-441b-8648-4c0361892ee0/blattella-germanica; Collembola icon, CC0 1.0, https://commons.wikimedia.org/wiki/File:Collembola_icon.svg; Zygentoma icon, CC0 1.0, https://commons.wikimedia.org/wiki/File:Zygentoma_icon.svg; Chrysopa, CC0 1.0, https://www.phylopic.org/images/5053ea6b-faf3-438a-82b1-b9b742eed5a0/chrysopa; Haploembia solieri, CC0 1.0, https://www.phylopic.org/images/82ea9749-7a98-4455-90c1-77cbaf2944bc/haploembia-solieri.

nestedness (nestedness-driven beta diversity (SNE) = 0.11; 13%) (Figs. 5D–5E). This is reflected in the fact that each trap has a high percentage of unique species, where the canopy fogging presents 65% of exclusive species, the pitfall 71%, and the Malaise 85%.

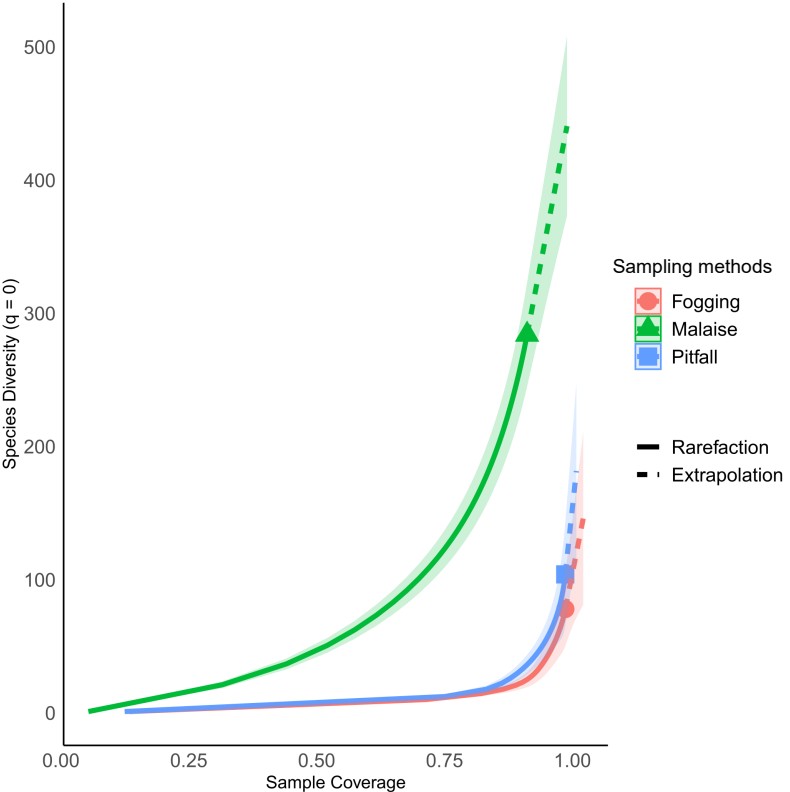

**Figure 3 Percentage of sampling coverage for each collection method.** Each line represents the dynamic behavior of the richness relative to the total sampling. Continuous line represents rarefaction, while dotted line represents extrapolation. Symbol indicates estimated richness. All three types of traps have an estimated coverage larger than 90% overall. Individual orders have different behavior (Fig. S1).

These results indicate that although methods such as pitfall traps or the canopy fogging method have lower species richness, they enable access to species composition that cannot be obtained by other methods.

When we observe the dissimilarity between paired samples, we can see that in general, all the samples share few elements (Fig. 5D, SOR = 0.86–0.89). However, the greatest distance in terms of composition is between the pitfall traps and the canopy fogging method, where there is a very low degree of nesting (nestedness) (SNE = 0.02). For its part, the flight interception traps, compared to the canopy fogging method had a greater number of nested or shared species, which shows us that although with specific elements, the Malaise trap can collect some of the elements collected by the fogging method (SNE = 0.21).

In terms of the representation of trophic guilds, we were able to identify the presence of 11 functional groups (Figs. 6–7). The Malaise traps were able to collect all guilds, while the pitfall and canopy fogging traps collected nine of them. As with species richness, the difference in trophic guilds for each collection method was also significant ($F = 31$, $P < 0.001$), with the Malaise trap having a greater ability to collect trophic guilds (Figs. 6–7).
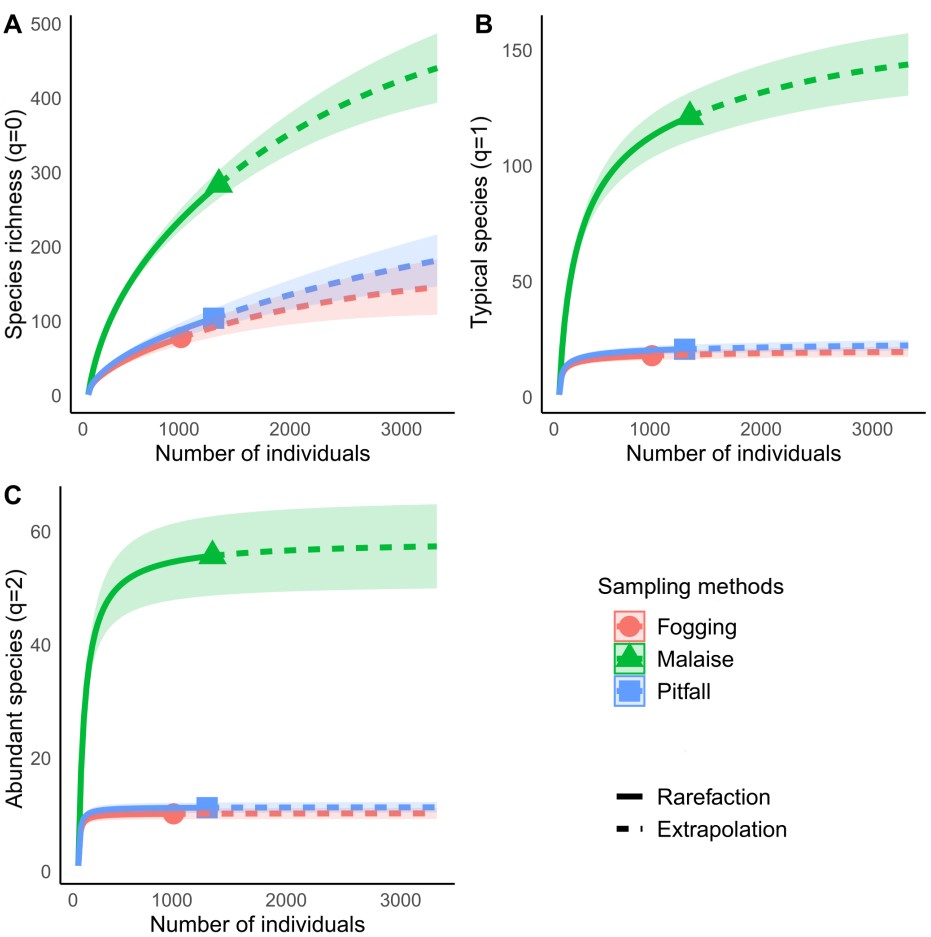

**Figure 4** **Rarefaction and extrapolation of diversity based on Hill numbers for Malaise, Fogging and Pitfall sampling methods.** (A) Species richness ($q = 0$), (B) Shannon Diversity, $q = 1$, (C) Simpson Diversity $q = 2$. Each curve is computed from 999 bootstraps. Continuous line indicates rarefaction. Dashed line indicates extrapolation. Symbol indicates estimated richness for each index. Shaded area represents 95% confidence intervals.

When we analyzed the affinity of each collection method for each trophic guild, considering the number of species, we observed that in general, the traps had a high evenness (J) with values greater than 80% (Table 1). This means that each collection method did not show dominance by a few trophic guilds. However, despite these results, we can see some trends worth highlighting. The Malaise traps had a greater representation of the saprophytic/saprophagous/coprophagous guilds (22%), parasitoids (21%), and predators (20%). Meanwhile, in the pitfall traps, the greater representation was given by the saprophytic/saprophagous/coprophagous (26%), omnivores (20%), and predators (16%). Finally, in the canopy fogging traps, the highest representation was given by the omnivorous (17%), parasitoid (16%), predatory (15%), and phytophagous sap-sucking guilds (14%) (Fig. 7).

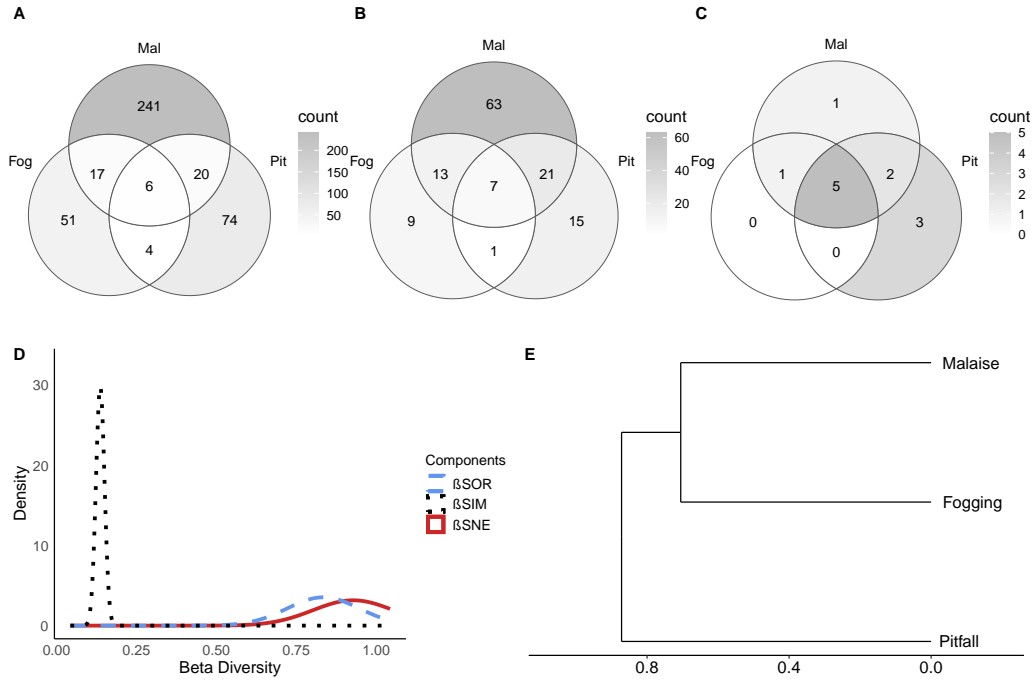

**Figure 5** **Beta diversity of insect communities with three collection methods evaluated in a relict of Dry Forest in Puerto Colombia.** (A–C) Venn diagrams of shared (A) morphospecies, (B) families and (B) orders among methods, respectively. Darker shades of gray indicate higher counts. (D) Partition of the diversity of βSOR (blue dashed line) between βSIM (black dotted line) and βSNE (red continuous line). (E) Clustering using the average of the βSIM dissimilarity of the species composition.

# DISCUSSION

We identified a rich diversity of insects in a relic of dry forest in Puerto Colombia, within the Metropolitan area of Barranquilla, with sampling coverage values exceeding 85%. Our collection methods not only complement each other, but also exhibit a high turnover of species. The Malaise traps were notably effective, securing more than double the species in comparison with the second most effective method, pitfall traps.

Despite the forest's size and context, it sustains various trophic guilds, indicating a valuable provision of ecosystem services offered by conserving such urban forest remnants. The entomofauna in our study area demonstrated a richness either equivalent or superior to those reported in other continental dry forests (*Janzen & Schoener, 1968*; *Janzen, 1973a*; *Janzen, 1973b*; *Vasconcellos et al., 2010*; *Macedo-Reis, Quesada & Neves, 2019*; *Silva et al., 2017*) and Caribbean islands (*Janzen & Schoener, 1968*; *Janzen, 1973a*; *Janzen, 1973b*). This richness, seen even in a limited area, highlights the conservation of a significant number of species and trophic guilds due to the presence of defined stratification. The diversity in terrestrial ecosystems correlates with tree abundance and is influenced by both structural complexity and plant species richness (*McKinney, 2008*; *Helden, Stamp & Leather, 2012*).

When our results are compared to larger-scale studies (*Janzen & Schoener, 1968*), those incorporating various time windows (*Janzen & Schoener, 1968*; *Silva et al., 2017*), or those

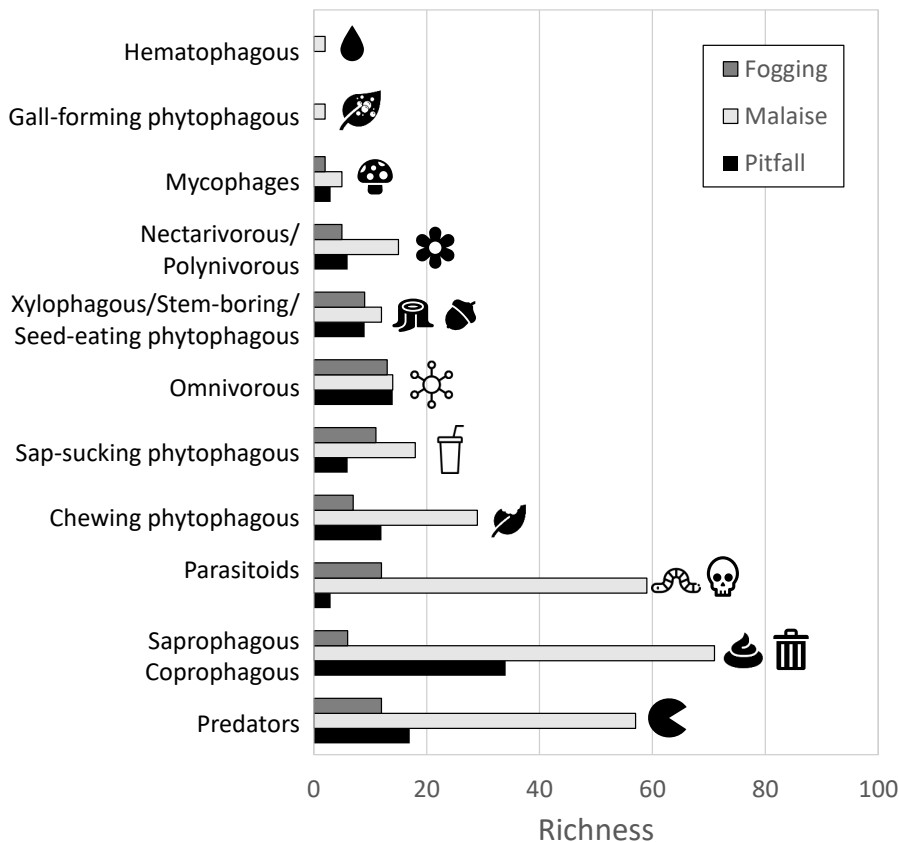

**Figure 6 Distribution of richness by detected guilds and collection methods.** Each bar represents the number of morphospecies belonging to a particular guild. Different colors represent different traps. Icons (source: Microsoft Office 365 license) are used to illustrate characteristics of each guild.

at the molecular scale (*Janzen et al., 2009*), the depicted diversity is but a fraction of what tropical dry forests can potentially harbor. Our data collection was temporally limited (wet season), and the reliance on morphological characters for morphospecies separation might have underestimated the richness within some species groups, particularly those with cryptic species (*Janzen et al., 2009*; *Yu et al., 2012*; *Geiger et al., 2016*).

It's crucial to acknowledge that our comparative global data derives from studies utilizing different collection methods or focusing on specific trophic guilds (*Janzen & Schoener, 1968*; *Janzen, 1973a*; *Janzen, 1973b*; *Vasconcellos et al., 2010*; *Macedo-Reis, Quesada & Neves, 2019*; *Silva et al., 2017*). There is a noticeable scarcity of broad-spectrum collection methods aimed at diverse taxa and trophic guilds in the Caribbean.

In our study, the Malaise trap demonstrated the highest efficiency in insect collection, supporting findings documented across various ecosystems (*Faulds & Crabtree, 1995*; *Darling & Packer, 1988*; *Shweta & Rajmohana, 2016*; *Sheikh et al., 2016*). The trap efficiently captures a wide range of terrestrial insects, collecting both ground-moving organisms and those flying at low levels (approximately 2.5 m) (*Marston, 1965*; *Sheikh et al., 2016*). It is also evident by the lack of saturation, that additional sampling efforts might yield higher

| Order | Total Richness | Pitfall | Malaise | Fogging |
|---|---|---|---|---|
| Saprophagous/Coprophagous | 100 | 6 ➡ | 71 ⬆ | 34 ➡ |
| Predators | 79 | 12 ➡ | 57 ⬆ | 17 ➡ |
| Parasitoids | 73 | 12 ➡ | 59 ⬆ | 3 ➡ |
| Chewing phytophagous | 43 | 7 ➡ | 29 ⬆ | 12 ➡ |
| Sap-sucking phytophagous | 32 | 11 ➡ | 18 ⬆ | 6 ➡ |
| Omnivorous | 31 | 13 ➡ | 14 ➡ | 14 ➡ |
| Xylophagous/Stem-boring Seed-eating phytophagous | 22 | 9 ➡ | 12 ⬆ | 9 ➡ |
| Nectarivorous/Polynivorous | 19 | 5 ➡ | 15 ⬆ | 6 ➡ |
| Mycophages | 9 | 2 ⬇ | 5 ➡ | 3 ➡ |
| Gall-forming phytophagous | 2 | ✖ | 2 ⬇ | ✖ |
| Hematophagous | 2 | ✖ | 2 ⬇ | ✖ |

|  | Species count | Sample Coverage | Total richness |
|---|---|---|---|
| ⬆ | > 10 | > 80 % | > 50 % |
| ➡ | ≥ 4 | > 50 % | > 30 % |
| ⬇ | < 3 | occasional sample | |
| ✖ | No reports | | |

**Figure 7 Richness by trophic guilds and efficiency of three different collection methods and their efficiency in a relict of dry forest in the municipality of Puerto Colombia–Atlántico.** Icons indicate efficiency and richness ranks for each guild and each sampling method. Icons (source: Microsoft Office 365 license) are used to illustrate characteristics of each guild.

richness, while the non-intersection with the other methods highlights the limitations of using a single sampling method (*Lamarre et al., 2012*).

The secondary dry forest studied here supports abundant undergrowth, which provides rich resources for insects. This is made possible by tree species adapted to dry conditions with smaller canopies, allowing sunlight to penetrate and promote undergrowth development (*Herazo Vitola, 2017*). This environment favors both undergrowth individuals and associated entomofauna during various successional stages.

The Malaise trap is not only adept at collecting a plethora of insect orders with special affinity for the most diverse ones—Coleoptera, Hymenoptera, and Diptera (*Triplehorn, Johnson & Borror, 2006*)—but also effectively doubles the diversity values reported by other methods.

Pitfall traps, while popular, inherently favor certain taxa, limiting their functional scope to ground-dwelling species like tenebrionids, staphylinids, scarabeids, formicids and blattids (*Montgomery et al., 2021*). Contrastingly, canopy fogging, a general inventory method with a wide taxonomic scale (*Basset et al., 1997*), yielded lower values than Malaise traps and did not significantly differ from pitfall traps. This outcome, surprising given previous studies (*Erwin, 1983*; *Adis, Lubin & Montgomery, 1984*), mirrored the richness

observed in the canopies of other continental dry forests (*Neves et al., 2010*; *Diodato & Fuster, 2016*), showcasing the variability in canopy habitats under different pressures or stresses (*Adis, Lubin & Montgomery, 1984*).

The Malaise trap excelled not only in species richness but also in abundance distribution across species ($q = 1$ and $q = 2$), whereas pitfall and fogging methods exhibited narrower affinities for select species. Such values may correlate with the habitat dynamics where collections occurred, given the forest's successional state offering a stable microclimate and abundant resources within the medium, and creeping strata where Malaise traps operate.

Furthermore, methodological differences were not confined to richness and evenness; there were distinct variations in species identity, underscoring the specificity of each method. This variance is likely due to the distribution of insect guilds across forest strata (*McKinney, 2008*; *Helden, Stamp & Leather, 2012*), as our intervened forest can host well-differentiated taxa layers.

Guild diversity, akin to taxonomic diversity, was unique for each method regarding both diversity measures and dominant guild identity per trap type. For instance, Malaise traps predominantly captured saprophytes/saprophages/coprophages due to the presence of dipterans (*Montgomery et al., 2021*), which, owing to their flight habits, rarely appear in pitfall traps. The traps also registered predators from the Hymenoptera, Coleoptera, and Diptera families, capitalizing on prey available on herbaceous and shrubby foliage (*Triplehorn, Johnson & Borror, 2006*), and parasitoids employing similar foraging techniques (*Shweta & Rajmohana, 2016*).

Pitfall traps primarily recorded saprophytic/saprophagous/coprophagous and omnivorous guilds, with decomposers mostly belonging to the Coleoptera order exploiting soil detritus (*Nichols et al., 2008*). Predatory guilds, represented by Coleoptera and Hymenoptera (Formicidae), utilized herbaceous vegetation and creeping lianas as major resources (*Janzen, 1973b*).

Lastly, canopy fogging revealed four trophic guilds with comparable richness values, all exploiting sprouts, flowers, or fruits in tree crowns, interacting with associated prey (*McKinney, 2008*; *Helden, Stamp & Leather, 2012*). This complex interaction of species and guilds underlines the intricate biodiversity sustained in these crucial urban forest remnants.

## CONCLUDING REMARKS

Our findings underscore the importance of employing multiple insect sampling methods to attain a comprehensive understanding of the insect diversity and community structure within TDFs. The unique attributes of each sampling technique are pivotal in capturing different insect taxa and trophic guilds, highlighting their complementary roles. The rich biodiversity found within the urban TDF of Puerto Colombia, even in the face of environmental stressors, emphasizes the necessity for concerted conservation efforts. This research not only contributes to the limited data on insect communities in the Caribbean TDFs, but also supports global initiatives aimed at biodiversity conservation in rapidly urbanizing environments.

## ACKNOWLEDGEMENTS

We thank Diethard Tautz for his scientific support in the establishment of this working group. We thank Cecilia Torres, Andres de la Hoz, and Anahi Barros for their help in the field and cleaning of samples in the laboratory. We thank Universidad del Norte for allowing us access to the forest. We hope projects like this will encourage the preservation of such areas to improve our understanding of life in the Tropics.

### Funding

This work was supported by Max Planck Partner Group funding to Rafik Neme. The funders had no role in study design, data collection and analysis, decision to publish, or preparation of the manuscript.

### Grant Disclosures

The following grant information was disclosed by the authors:
Max Planck Partner Group.

### Competing Interests

The authors declare there are no competing interests.

### Author Contributions

- Rodrigo Sarmiento-Garcés conceived and designed the experiments, performed the experiments, analyzed the data, prepared figures and/or tables, authored or reviewed drafts of the article, and approved the final draft.
- Daniel Posada-Echeverría analyzed the data, prepared figures and/or tables, and approved the final draft.
- Rafik Neme conceived and designed the experiments, analyzed the data, prepared figures and/or tables, authored or reviewed drafts of the article, and approved the final draft.

### Data Availability

The morphospecies, their taxonomic placement, trophic guilds, distribution in traps and abundances are available in the Supplemental Files.

### Supplemental Information

Supplemental information for this article can be found online at http://dx.doi.org/10.7717/peerj.18262#supplemental-information.

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
