# Peer review of "Diversity and efficacy of insect sampling methods in an urban tropical dry forest of the Colombian Caribbean"

_PeerJ, doi:10.7717/peerj.18262_

## Round 0.1 · original submission · Minor Revisions

All three reviewers made interesting suggestions about the study, while recognising its value. I encourage you to follow them to improve the paper.

Reviewer 1 ·

Basic reporting

Observations on the text (see also the PDF attached)
Page 22 “in regions with tropical climate” maybe be redundant, there is not TDF in temperate climates.
26 altitude(González-M - split
28 nutrient cycling, and economic and recreational values (ad and).
30 delete comma in “… change, and …”
42 IAVH 1998 not in alphabetical order in references, also change IAVH by IAvH
50 and ss. It is true, but in addition to Malaise traps, comprehensive studies are also needed for other types of traps such as Winkler or pitfall.
52 Be careful with the use of commas in references, standardize
57 Why not Winkler traps? These prove to be efficient in sampling arthropod in soil and litter
74 IDEAM not in references
78 Perhaps is better to refer to “secondary tropical dry forest”, because no original forest survives the human intervention
84 Why weren't Winkler traps used? These traps have proven to be very useful for soil and leaf litter arthropod fauna
93 Why was sampling not done in the dry season? This would provide data from two contrasting seasons (rainy and dry), in addition to evaluating the effectiveness of the traps in different seasons.
99 Eight days of time for pitfall traps can be a long time, in which time samples can dry out and deteriorate. How did you prevent rain from filling the glasses and losing some samples?
104 It would have been interesting to place Malaise traps at the edge of the forest. The ecotones keep a good diversity of arthropods
127 – 128 change “y” by “&”
144 Data Analysis. Since I am not an ecologist, I cannot comment on this section.
186 Why not (Figures 1 & 2)?
191 as above
205 – 206, 210 -211 observation as in 186
223 Cassidinae is subfamily of Chrysomelidae
224 correct “Curcuionidae”
225 “Homoptera” is order name (not natural), not family name
226 correct “Ploiarridae”
227 put Pscoptera as “Psocoptera” as this order is paraphyletic
229 Acanthopodiae – fix name
230 Bolboceratinae is subfamily
232 correct “Mutilidae”
233 correct “Scarabeidae”
235 is families, not family
236 Apoidea is superfamily
237 Bradynobaenidae currently is absent of northern Neotropics, must be Cyphotidae
237 Cassidinae is subfamily
238 the correct name is Chamaemyiidae
239 Chrysidoidea is superfamily name
259 and ss Remember that fogging does not sample all potential fauna because some insects remain attached to the vegetation and do not fall into the collectors.
288 In comparison with …
297, 298, 304, 305, 313 Janzen incomplete autor citation
222 and ss The comparison of the 51 species of families and the “high diversity of Psocoptera” is unclear. What is diversity like in non-Psocoptera?
307 In addition to the lack of dry season sampling there is the issue that other traps (such as Winkler) could trap entomofauna that the other traps do not capture. This can also be noted for other traps such as light, baits or even manual capture
324 what means “secondary environments”?
335 better: Scarabeids, formicids and blattids
363 Nicholls et al not in references
450 Change Fernandéz by: Fernández
466 Carrejo(1992) split
473 – 474 Separate space between references
518 Reference missplaced
576 follow the autor citation guide
581 Correctly cite the publication
584 M by Michael
592 fiel guide by: Field Guide
Figures 1 and 2 Change Mantoidea by Mantodea, put Psocoptera as “Psocoptera”
Figure 5 relict of a urban Dry Forest (add urban)

Experimental design

no comment in this section

Validity of the findings

no comment

Additional comments

Tropical dry forests are certainly one of the most threatened ecosystems in the world. In Colombia there are very few intact areas of this ecosystem, with few conservation options. This is a tragedy for biodiversity conservation since several studies suggest that these ecosystems include a high diversity of species, and possible endemism, in several groups of species such as bees or longhorn beetles. Therefore, any contribution that explores insect diversity in three different areas as shown in this submitted manuscript is welcome.

My observations are necessarily limited as I am not an ecologist and therefore not an expert in statistical parameters and analysis used by the authors.

First of all, although from the beginning we talk about tropical dry forest (TDF), since these are forests in urban environments, I do not know how much the term “dry forest” applies when we are talking about urban environments, where there are not necessarily patches continuous forests of certain sizes that guarantee the term forest. Rather, they are fragments of forest. There is no problem in studying the fauna of urban environments, since these environments can eventually be refuges for threatened species and that support very altered conditions such as urban or semi urban places.
Authors must be careful in the use of commas and semicolons in citations, there are inconsistencies, some detected and pointed out. For example, in some cases the author's initial is placed and in others the full name.

Overall rating. There are few areas of tropical forests left in Colombia and probably in other regions. It is an ecosystem rich in species and ecosystem services, but highly threatened, which is why any study of its taxonomic and functional diversity (guilds) is welcome, as it offers key data for its protection, conservation and monitoring. This study could extend in time (other seasons), traps (e.g. Winkler) and taxonomic detail (at least genera). Taxonomically well-known groups (such as ants) could be used for eventual monitoring.

Annotated reviews are not available for download in order to protect the identity of reviewers who chose to remain anonymous.

·

Basic reporting

The manuscript is written in clear language and uses professional English throughout. The language is technically correct and adheres to professional standards of courtesy and expression. There are no significant grammatical or typographical errors that impede comprehension.
The introduction and background sections provide a thorough context for the study. The authors reference relevant literature extensively, including key studies on tropical dry forests, insect sampling methods, and the ecological roles of insects. This establishes a solid foundation for understanding the significance of the research.
The manuscript conforms to the professional structure expected in scientific articles. It includes well-organized sections such as the introduction, methods, results, discussion, and conclusion. Figures and tables are relevant, high-quality, well-labeled, and described in detail. The raw data is supplied in accordance with PeerJ's policy, ensuring transparency and reproducibility.

Experimental design

The study represents original primary research that falls within the aims and scope of the journal.
It addresses a significant ecological question regarding the efficacy of different insect sampling methods in a specific and understudied ecosystem.
The research question is well-defined, relevant, and meaningful for conversation efforts. The authors clearly state how their research fills an identified knowledge gap, particularly in the context of insect diversity in tropical dry forests, which are among the most threatened ecosystems globally.
The methods are described but lag some crucial details and information that are needed for reproductivity.

Validity of the findings

While the impact and novelty of the findings are not explicitly assessed, the study's contribution to the literature is evident. The use of multiple sampling methods and the comprehensive data set provide valuable insights into insect diversity in tropical dry forests.
The authors provide all underlying data, which seem robust, statistically sound underlining the findings.
However, I would like more explanation of the usage of the iNEXT package to argue the saturation of the sampling depth. The measurements of q1 and q2 as shown in the MS are somewhat confusing since there is no deeper ecological question asked before. One problem I have with using iNEXT is that the Inverse Simpson (q2) in theory could never be saturated stated by the authors of the package and the shown values around 8 for pitfall and fogging stand against the maximum value for the Malaise traps with a value around 50.
Besides this minor problem, I fail to understand (and I looked through all the documentation) how multiple Malaise traps would not pick up any Lepidoptera? Same question for spiders in the pitfall traps. I cannot believe that these orders of arthropods were not present at all. I think the authors need to address that in their sample design and discussion.

However, the conclusions are well-stated and directly linked to the original research question. They are appropriately limited to the supporting results as the MS stands right now which provides a clear and concise summary of the study's findings and their implications for conservation efforts in tropical dry forests.

Additional comments

Minor comments and suggestions can be found in the .pdf file

Reviewer 3 ·

Basic reporting

The text of the article was well written with appropriate references and proper structure. Only the figure captions in the text were sometimes incorrect. Table 1 was also incorrectly labelled as Table 3 in the table caption.
I also think that figure 2 does not give any additional information that is not already given in figure 1. Also there might be a simpler way of presenting the data, a stacked bar chart with additional richness numbers for each insect order might be an option. Figure 3 should be explained a bit more and could also be included in Figure 4. And it would be more intuitive to change the labels for the figures generated by the iNext package, as the sampling methods are labelled as Assemblage and q=0/1/2 could be replaced directly in the figures with their alpha diversity index used.

Experimental design

It is extremely important to generate data from different environments, especially those that are at risk of being destroyed by human impact, climate change or other stressors, so that conservation measures can be identified. So it makes sense to study urban tropical dry forests and perhaps compare their importance and diversity with other sites or in the forests themselves, generating information about different insect clusters in the different layers of the forest. However, I have to admit that comparing different traditional sampling approaches is hardly new. So if it is possible, perhaps the focus could be a little different.
The research question was mainly to compare sampling methods, but also to assess the diversity in the TDF. This could be defined a bit more clearly, as the comparison of methods alone does not bring any new information, so perhaps it would be preferable to focus more on the data evaluated and the insect communities found in the forest (perhaps in comparison to other findings) and also at different strata of the forest.
The methods used and the procedure were nicely described.

Validity of the findings

no comment

Additional comments

L204: "insect community"
I would use a different word, because the malaise trap, as you say, does not cover whole communities, it is missing a lot of ground dwelling insects in particular. But I agree that it is the best method to catch the highest number of insects.

---

## Round 0.2 · accepted · Accept

You have addressed appropriately the comments from the reviewers and revised the manuscript accordingly when it was possible or necessary to do it. I thus consider the ms acceptable for publication.